

# Genome-wide identification and analysis of the thiolase family in insects

Shou-Min Fang

Key Laboratory of Southwest China Wildlife Resources Conservation (Ministry of Education), China West Normal University, Nanchong, Sichuan, China
College of Life Science, China West Normal University, Nanchong, Sichuan, China

## ABSTRACT

Thiolases are important enzymes involved in lipid metabolism in both prokaryotes and eukaryotes, and are essential for a range of metabolic pathways, while, little is known for this important family in insects. To shed light on the evolutionary models and functional diversities of the thiolase family, 137 thiolase genes were identified in 20 representative insect genomes. They were mainly classified into five classes, namely cytosolic thiolase (CT-thiolase), T1-thiolase, T2-thiolase, trifunctional enzyme thiolase (TFE-thiolase), and sterol carrier protein 2 thiolase (SCP2-thiolase). The intron number and exon/intron structures of the thiolase genes reserve large diversification. Subcellular localization prediction indicated that all the thiolase proteins were mitochondrial, cytosolic, or peroxisomal enzymes. Four highly conserved sequence fingerprints were found in the insect thiolase proteins, including CxS-, NEAF-, GHP-, and CxGGGxG-motifs. Homology modeling indicated that insect thiolases share similar 3D structures with mammals, fishes, and microorganisms. In *Bombyx mori*, microarray data and reverse transcription-polymerase chain reaction (RT-PCR) analysis suggested that some thiolases might be involved in steroid metabolism, juvenile hormone (JH), and sex pheromone biosynthesis pathways. In general, sequence and structural characteristics were relatively conserved among insects, bacteria and vertebrates, while different classes of thiolases might have differentiation in specific functions and physiological processes. These results will provide an important foundation for future functional validation of insect thiolases.

## INTRODUCTION

Thiolases are ubiquitous enzymes that play important roles in lipid-metabolizing pathways (*Thompson et al., 1989*; *Igual et al., 1992*; *Pereto, Lopez-Garcia & Moreira, 2005*). There are two major kinds of thiolases based on the direction of the catalytic reaction (*Masamune et al., 1989*; *Modis & Wierenga, 2000*). One is degradative thiolase I (3-ketoacyl-CoA thiolase, E.C. 2.3.1.16), which catalyzes the thiolytic cleavage of medium- to long-chain unbranched 3-oxoacyl-CoAs (from 4 to 22 carbons) into acetyl-CoA and a fatty acyl-CoA (Fig. S1A) (*Clinkenbeard et al., 1973*; *Schiedl et al., 2004*; *Houten & Wanders, 2010*). It is mainly involved in fatty acid $\beta$-oxidation and preferentially catalyzes the last step. The other is biosynthetic thiolase II (acetoacetyl-CoA thiolase, EC2.3.1.9), which is capable of

Corresponding author
Shou-Min Fang,
fangshoumin@126.com

catalyzing the Claisen condensation reaction of two molecules of acetyl-CoA to acetoacetyl-CoA (Fig. S1B). Thiolase II might be involved in poly beta-hydroxybutyric acid synthesis, steroid biogenesis, etc. (*Clinkenbeard et al., 1973*).

Based on the function, oligomeric state, substrate specificity, and subcellular localization, six different classes of thiolases (CT, AB, SCP2, T2, T1, and TFE) have been identified in humans (*Fukao, 2002*; *Mazet et al., 2011*; *Anbazhagan et al., 2014*). In trypanosomatid and bacterial kingdoms, another four classes were identified, including thiolase-like protein (TLP), SCP2-thiolase-like protein (SLP), unclassified thiolase (UCT), and TFE-like thiolase (TFEL) (*Mazet et al., 2011*; *Anbazhagan et al., 2014*). The CT-thiolase, located in the cytosol, has a key role in catalyzing the condensation of two molecules of acetyl-CoA to acetoacetyl-CoA, which is the first reaction of the metabolic pathway leading to the synthesis of cholesterol (*Kursula et al., 2005*). AB-thiolase and SCP2-thiolase as degradative thiolases occur in peroxisomes (*Antonenkov et al., 1997*; *Antonenkov et al., 1999*). T1-, T2-, and TFE-thiolases are mitochondrial degradative enzymes. Except for the degradation of acetoacetyl-CoA and 2-methyl-acetoacetyl-CoA, T2-thiolase has a biosynthetic function in the synthesis of acetoacetyl-CoA in ketone body metabolism (*Fukao et al., 1997*). In general, AB-, SCP2-, T1-, and TFE-thiolases belong to degradative thiolase I, and CT class is biosynthetic thiolase II. Importantly, T2-thiolase is a bi-functional enzyme with synthesis and degradation activity.

As enzymes responsible for broad pathways, thiolases have been performed the phylogenetic analysis in mycobacteria and functional studies in humans (*Mazet et al., 2011*; *Xia et al., 2019*). In insects, several thiolase genes have been characterized (*Fujii et al., 2010*). In *Helicoverpa armigera*, an acetoacetyl-CoA thiolase was cloned and performed functional analysis (*Zhang et al., 2017*). It was indicated that the thiolase was involved in the early step of the juvenile hormone pathway, i.e., mevalonate biosynthesis. One acetoacetyl-CoA thiolase was purified to apparent homogeneity by column chromatography in *Bombus terrestris*, suggesting that it might be the first enzyme in the biosynthesis of terpenic sex pheromone (*Brabcova et al., 2015*). Besides, acetyl-CoA can be used as the precursor for *de novo* biosynthesis of sex pheromones in female moths, and four 3-ketoacyl-CoA thiolase genes were identified in sex pheromone gland transcriptome of the noctuid moth *Heliothis virescens* (*Vogel et al., 2010*). In total, there are few and scattered studies on insect thiolase, lacking systematic identification and comparative studies.

In this study, we selected 20 representative species from seven insect orders to perform genome-wide identification of the thiolase family proteins. Gene structure, chromosome location, and three-dimensional (3D) structure and motif characteristics of proteins were compared. In addition, *Bombyx mori* is an important model species for studying juvenile hormone and sex pheromone biosynthesis (*Matsumoto, 2010*; *Xia, Li & Feng, 2014*). Expression profiles of the thiolase genes were detected in various tissues and developmental sex pheromone gland of *B. mori*. Combining structural characteristics and expression patterns, the potential functions and involved physiological processes were hypothesized. The present study can help us understand the functional differentiation of thiolase genes in insects.

**Table 1  Classification of the thiolase genes among insects, *H. sapiens* and *M. tuberculosis*.**

| Organisms | Total | CT | T1 | T2 | AB | TFE | TFEL (type-1) | TFEL (type-2) | SCP2 (type-1) | SCP2 (type-2) |
|---|---|---|---|---|---|---|---|---|---|---|
| *H. sapiens* | 6 | 1 | 1 | 1 | 1 | 1 | | | 1 | |
| *M. tuberculosis* | 9 | | | 1 | | 1 | 3 | 1 | 1 | 1 |
| Lep_*B. mori* | 6 | | 3 | 1 | | 1 | | | 1 | |
| Lep_*P. xuthus* | 10 | | 2 | 2 | 2 | 1 | | | 3 | |
| Lep_*M. sexta* | 6 | | 3 | 1 | | 1 | | | 1 | |
| Lep_*D. plexippus* | 5 | | 2 | 1 | | 1 | | | 1 | |
| Lep_*H. melpomene* | 4 | | 1 | 1 | | 1 | | | 1 | |
| Lep_*P. xylostella* | 15 | 1 | 10 | 1 | | 2 | | | 1 | |
| Dip_*C. quinquefasciatus* | 6 | 2 | 1 | 1 | | 1 | | | 1 | |
| Dip_*A. gambiae* | 7 | 1 | 2 | 1 | | 2 | | | 1 | |
| Dip_*D. melanogaster* | 6 | 1 | 1 | 1 | | 1 | | | 2 | |
| Col_*A. glabripennis* | 5 | | 2 | 2 | | | | | 1 | |
| Col_*N. vespilloides* | 6 | 1 | 1 | 2 | | 1 | | | 1 | |
| Col_*T. castaneum* | 5 | 1 | 1 | 1 | | 1 | | | 1 | |
| Hem_*H. halys* | 5 | 1 | 1 | 1 | | 1 | | | 1 | |
| Hem_*A. pisum* | 12 | 7 | 1 | 1 | | 1 | | | 2 | |
| Hem_*C. lectularius* | 8 | 2 | 1 | 1 | | 2 | | 1 | 1 | |
| Hym_*D. alloeum* | 8 | 2 | 1 | 3 | | 1 | | | 1 | |
| Hym_*A. mellifera* | 5 | 1 | 1 | 1 | | 1 | | | 1 | |
| Hym_*B. impatiens* | 8 | 1 | 2 | 2 | | 1 | | | 2 | |
| Pht_*P. humanus* | 6 | 1 | 1 | 2 | | 1 | | | 1 | |
| Iso_*Z. nevadensis* | 4 | 1 | 1 | 1 | | 1 | | | | |

**Notes.**

Lep, Lepidoptera; Dip, Diptera; Col, Coleoptera; Hem, Hemipter; Hym, Hymenoptera; Pht, Phthiraptera; Iso, Isoptera.

Thiolase-like protein (TLP) of *M. tuberculosis* was not listed in the table, and was not found homologous genes in insects and human.

# MATERIALS & METHODS

## Data resources

In this study, 20 representative species were selected from Lepidoptera, Hymenoptera, Hemiptera, Diptera, Coleoptera, Phthiraptera, and Isoptera (Table 1). The annotated genes and genomes of *B. mori* were retrieved from SilkDB v3.0 (https://silkdb.bioinfotoolkits.net). The sequence information of *Danaus plexippus* and *Heliconius melpomene* were downloaded from http://metazoa.ensembl.org/. *Manduca sexta* was downloaded from https://i5k.nal.usda.gov/. The other sequences were retrieved from GenBank (https://www.ncbi.nlm.nih.gov/), including *Papilio xuthus*, *Plutella xylostella*, *Culex quinquefasciatus*, *Anopheles gambiae*, *Drosophila melanogaster*, *Anoplophora glabripennis*, *Nicrophorus vespilloides*, *Tribolium castaneum*, *Halyomorpha halys*, *Acyrthosiphon pisum*, *Cimex lectularius*, *Diachasma alloeum*, *Apis mellifera*, *Bombus impatiens*, *Pediculus humanus*, and *Zootermopsis nevadensis*.

## Identification of insect thiolase genes

The known thiolase sequences of *Homo sapiens* and *Mycobacterium tuberculosis* were retrieved from GenBank (Table S1; *Anbazhagan et al., 2014*) and used as queries to perform BLASTP search (*E*- value <0.01) against the protein database of predicted genes in each species. Hidden Markov Model (HMM) files of Thiolase_N (PF00108) and Thiolase_C (PF02803) domains were downloaded from Pfam database (http://pfam.xfam.org/), which were used to screen the protein database of each species with hmmsearch in HMMER 3.0 (*E*- value <0.01). Based on BLASTP and hmmsearch analyses, the candidate thiolase genes were identified and subsequently checked by conserved domain search (CD-Search) in NCBI and hmmscan against Pfam database (*E*- value <1e−5). The candidate sequences that have Thiolase_N and/or Thiolase_C domains were recognized as thiolases. Those identified thiolases were used as new queries to perform BLASTP search against the protein database of each species until no more novel loci can be found. All the validated thiolase genes were used for further analysis.

## Phylogenetic analysis

The protein sequences of thiolases from 20 insects, *H. sapiens* and *M. tuberculosis* were aligned using MUSCLE (*Edgar, 2004*). Positions that had a high percentage of gaps (>70%) were trimmed. The handled alignment of protein sequences was used for checking the most suitable model of evolution by ProtTest 3.2 (*Darriba et al., 2011*). Maximum-likelihood (ML) trees were reconstructed using RAxML version 8.2.12 (*Stamatakis, 2014*) with the most suitable model (PROTGAMMAVTF) and 500 bootstrap replicates. FigTree v1.4.3 (http://tree.bio.ed.ac.uk/software/figtree/) was used for plotting the final phylogenetic tree. The clustering and classification of the thiolase sequences in the ML tree were done using known functional properties of *H. sapiens* and *M. tuberculosis* (*Anbazhagan et al., 2014*).

## Chromosome distribution, gene structure, and syntenic analysis

To localize the thiolase genes on chromosomes, *B. mori*, *H. melpomene*, *D. melanogaster*, *T. castaneum*, and *A. mellifera* were selected because their genome sequences have been assembled into chromosomes. Based on the GFF (General Feature Format) file of each species, every thiolase gene was mapped to the corresponding chromosomes. Using protein sequences of the thiolases, the precise exon/intron structures were generated through BLAT search against the genome sequences with Scipio server (https://www.webscipio.org/). The synteny events between two species were detected by Multiple Collinearity Scan toolkit (MCScanX) with the default parameters (*Wang et al., 2012*). The syntenic map of *B. mori* and *H. melpomene* was constructed with family_circle_plotter.java in MCScanX software.

## Molecular modeling of protein structure

The three-dimensional (3D) structure prediction of insect thiolases was conducted using the homology modeling method. Structures of T1-, T2-, CT-, AB-, TFE-, and SCP2-thiolases were predicted on-line at the SWISS-MODEL Interactive Workspace (*Arnold et al., 2006*). The known protein that has the highest sequence similarity to the thiolase to be analyzed is used for homology modeling. The predicted models of monomer and multimer were visualized in Swiss-PdbViewer 4.1.0 (*Guex & Peitsch, 1997*). To understand the 3D

structural similarities among the insect thiolases, all the other structures were compared with BmorT2 using magic fit algorithm in Swiss-PdbViewer, respectively. The root-mean-square distance (RMSD) values were calculated to express the structural similarity. The lower value of RMSD means higher similarity between two structures (*Carugo & Pongor, 2001*).

### Reverse transcription-polymerase chain reaction (RT-PCR)

The various tissues on day 3 of fifth-instar larvae were dissected in the silkworm. The sex pheromone glands (PGs) from five individuals were used as one sample at each developmental stage. All the samples were preserved in RNAlater (Ambion, 98 Austin, USA) and stored at −80 °C for RNA isolation. Total RNA was extracted using Trizol reagent (Invitrogen, USA). The first strand of cDNA was synthesized by M-MLV reverse transcriptase following the manufacturer's instructions (Promega, USA). RT-PCR primers were listed in Table S2. The silkworm *RpL3* gene was used as an internal control for relative quantitative analysis of RT-PCR. PCRs were performed with the following cycling parameters: 95 °C for 3 min (min), followed by 25 cycles of 30 s (s) at 95 °C, 30 s annealing (temperatures listed in Table S2) and 30 s extension (72 °C), and a final extension at 72 °C for 10 min. The amplification products were monitored on 1.5% agarose gels.

## RESULTS

### Genome-wide identification and phylogeny of insect thiolase proteins

To identify thiolases in insects, human and *M. tuberculosis* thiolase protein sequences were used as queries to perform homologous searches in whole genomes. In total, 137 thiolase genes were identified in 20 insects from seven orders, and the gene numbers of the species were ranged from 4 to 15 (Table 1; Table S1). The thiolase protein sequences were used to reconstruct the maximum-likelihood phylogenetic tree (Fig. 1). Based on the nomenclature rules in humans and *M. tuberculosis* (*Anbazhagan et al., 2014*), each thiolase was named in insects. It was indicated that insect thiolases were grouped 7 classes, namely CT, T1, T2, TFE, SCP2 (type-1), TFEL (type-2), and AB (Table 1; Fig. 1). Relatively, the classification of insect thiolases was more similar to that of the human than *M. tuberculosis*. Unlike humans, out of 20 insect species, only *P. xuthus* has gene members in AB class. Interestingly, 5 out of 6 Lepidopteran insects have no CT-thiolase. Furthermore, TFEL (type-2) class was only detected in *C. lectularius* (Hemipter) and bacterium *M. tuberculosis*.

To understand the evolutionary mode, gene gain and loss of thiolases were analyzed. It was indicated that most of the gain and loss events were occurred in a certain species (Fig. 2). Especially, *P. xylostella* (Lepidoptera), *A. pisum* (Hemipter), and *P. xuthus* (Lepidoptera) showed more duplications after or during the formation of the species, resulting in a total number of 15, 12, and 10 genes, respectively. Except for the gene duplication of a single species lineage, the common ancestor of *D. alloeum*, *A. mellifera* and *B. impatiens* showed 2 duplications (Fig. 2). This phenomenon was also noted in the clade of Hemipteran *H. halys*, *C. lectularius*, and *A. pisum*. For those recent duplication genes, they were often phylogenetically closely related to its ancestral genes (Fig. 1). Conversely, *A. mellifera* (Hymenoptera), *H. halys* (Hemipter), and *Z. nevadensis* (Isoptera) presented 3, 3, and 2

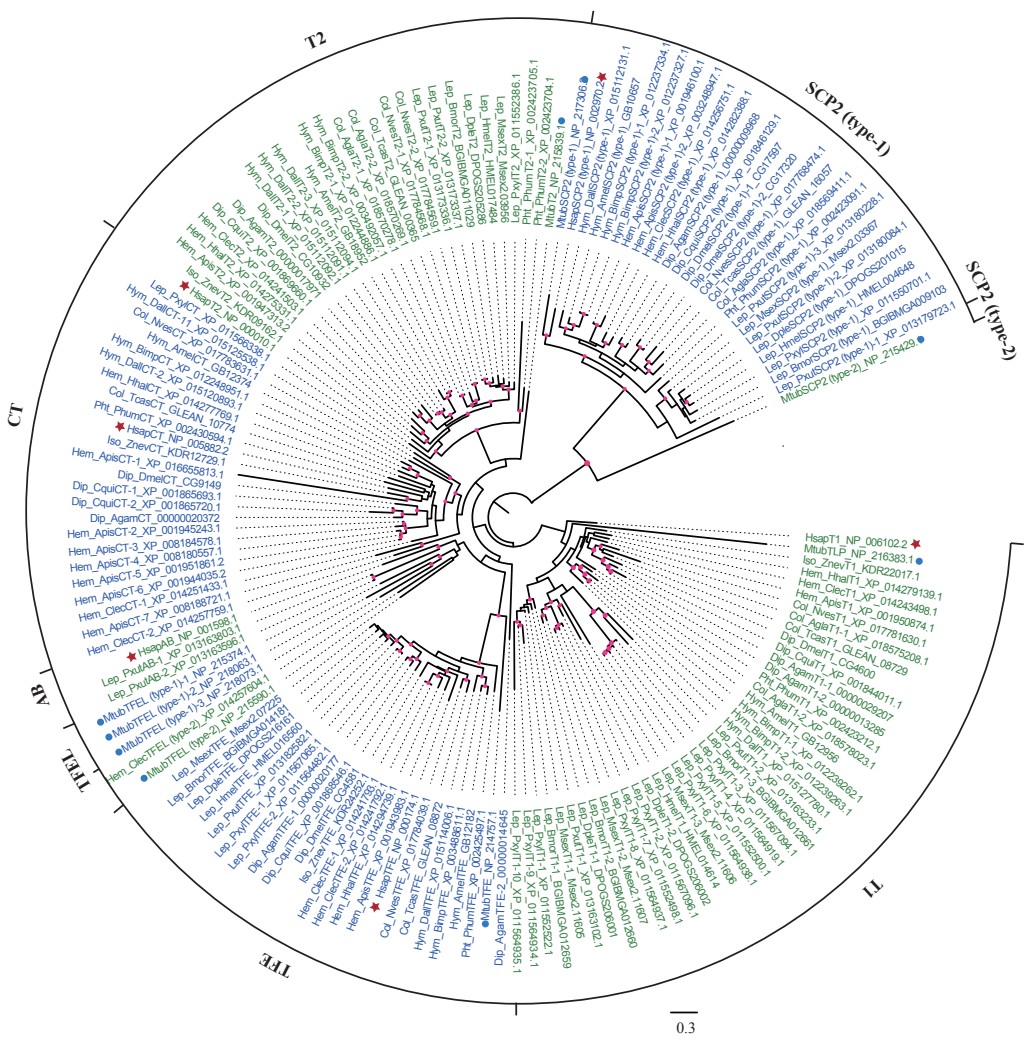

**Figure 1** **Phylogenetic tree of insect thiolases using the maximum-likelihood (ML) method.** The thiolases of human and *M. tuberculosis* were shown by stars and dots, respectively. The bootstrap values higher than 50% were dotted on the nodes. Lep: Lepidoptera; Dip: Diptera; Col: Coleoptera; Hem: Hemipter; Hym: Hymenoptera; Pht: Phthiraptera; Iso: Isoptera. The accession numbers following each gene name were presented. The information ENSANGP within accession numbers of *A. gambiae* was omitted.

gene losses during speciation, resulting in fewer genes in these species. Generally, gene gain and loss rates are important for understanding the role of natural selection and adaptation in shaping gene family sizes. For the species with more gene expansion, whether these duplicated genes play roles in adapting to special habitats deserves further study.

## Gene structures of insect thiolase genes

A comparative analysis of exon-intron structures was conducted for the 137 insect thiolase genes (Figs. 3A and 3B; Fig. S2A). The insect thiolase genes have a different number of introns ranging from 0 to 22. It was indicated that only 12 genes have no intron, and 17

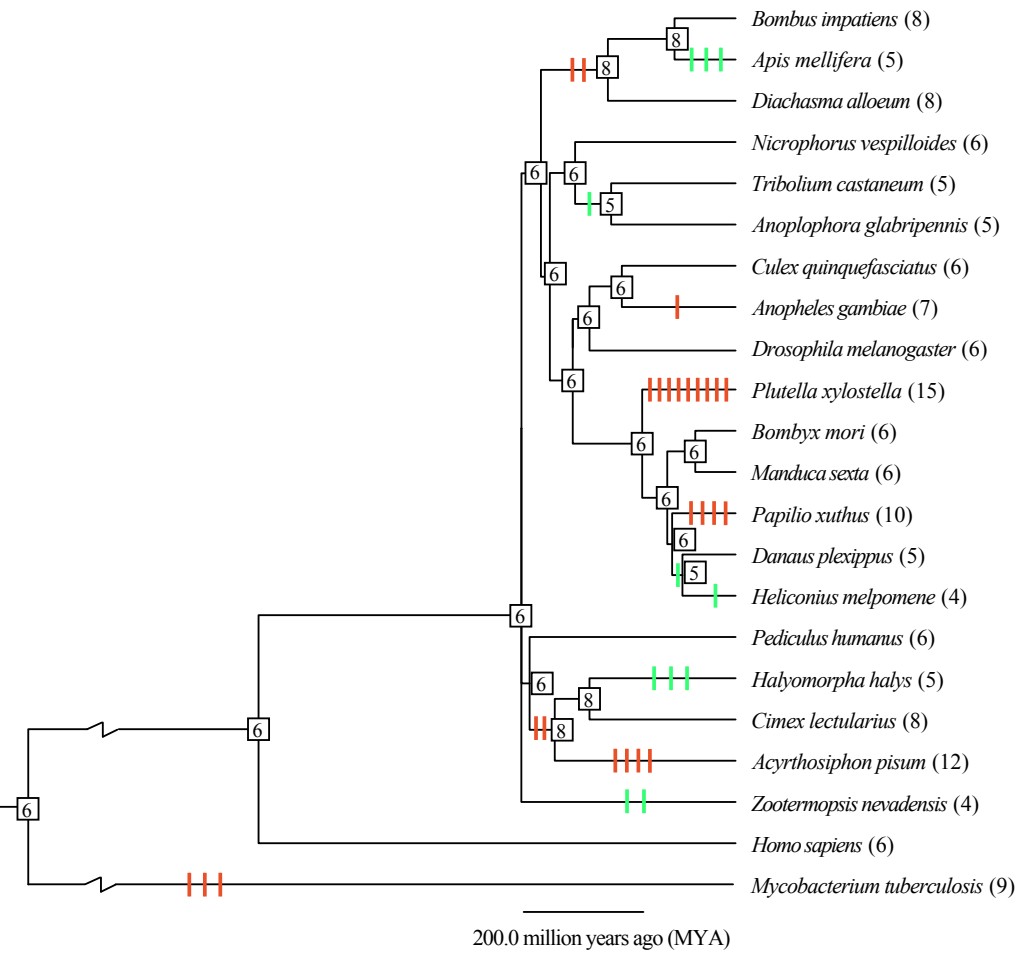

**Figure 2** **Gene gain and loss analysis of the thiolase gene family in insect genomes.** The species tree was obtained from the TimeTree database (http://www.timetree.org/). Gain and loss analysis was conducted by Notung-2.9 software with default parameters. The orange and green vertical bars on branch presented gene gain and loss, respectively. The number in each node is the gene count. The gene number of each species was presented in brackets.

genes have only one intron, accounting for 21.17% in total. The intronless genes were distributed in T2 (5), SCP2 (type-1) (2), CT (2), AB (2), and TFEL (1). Previous studies revealed that introns can delay regulatory response and are selected against in genes whose transcripts need to be adjusted quickly to meet environmental challenges (*Jeffares, Penkett & Bahler, 2008*). The intronless thiolase genes and the genes contained fewer introns might play important roles in survival for environmental changes. In addition, the intron number and exon/intron structures of thiolase genes are very different, even the orthologous genes of different species in the same class have a large differentiation. It was suggested that the differentiation of the intron number may result in the diversification of thiolase gene structures in insects.
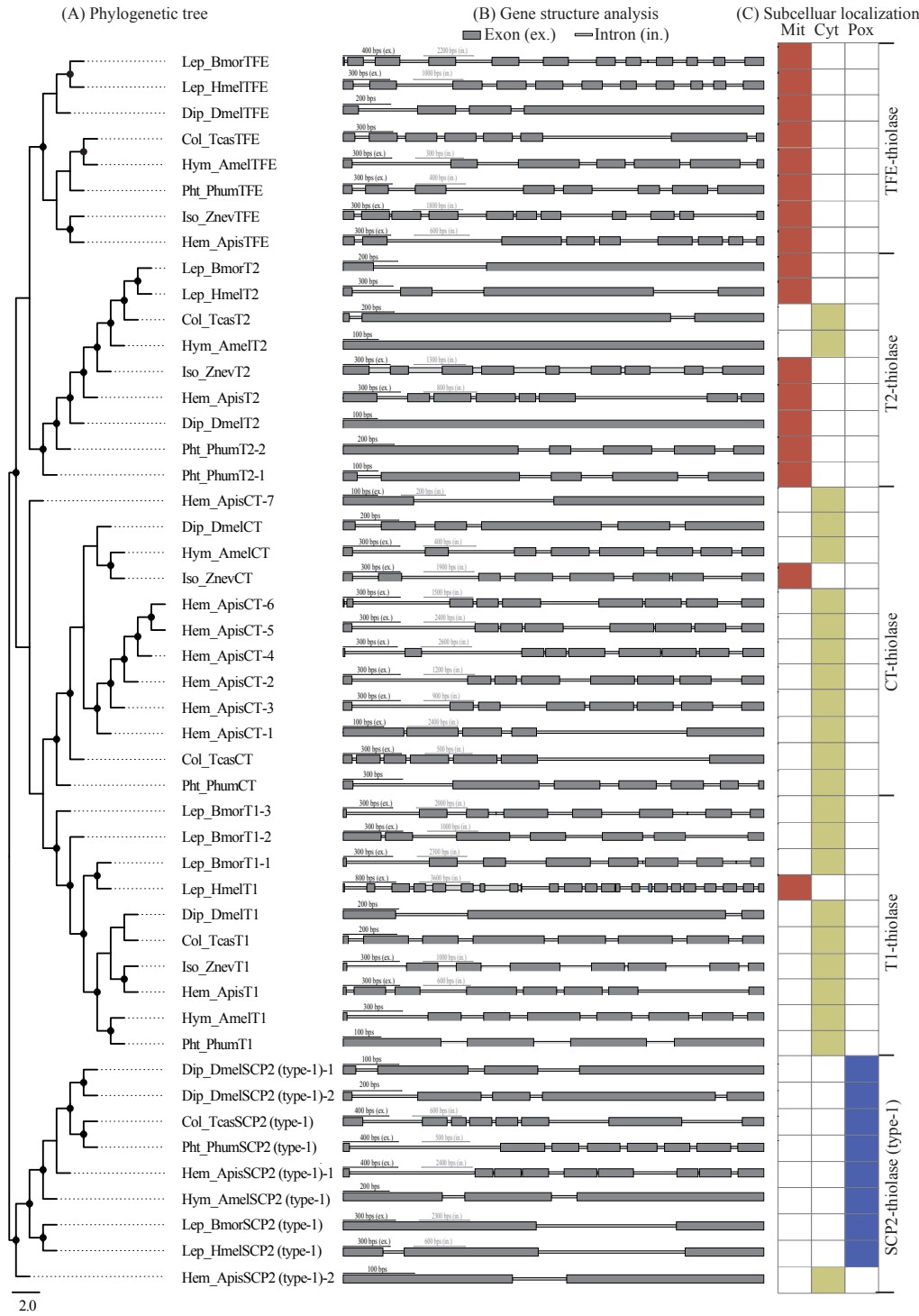

**Figure 3  Exon/intron structure and subcelluar localization analyses.** (A) The maximum-likelihood phylogenetic tree of thiolase proteins from seven representative insects. The bootstrap values higher than 50% were dotted on the nodes. (B) Exon/intron structure analysis of the thiolase genes. (C) Subcelluar localization analysis of thiolase proteins. It was predicted by the PSORT II server (https://www.genscript.com/tools/psort). Mit, mitochondrion; Cyt, cytosol; Pox, peroxisome.

## Chromosome distribution and gene synteny

In order to explore the chromosomal distribution of thiolase genes, five representative species were analyzed. It was indicated that most of the thiolase genes were randomly distributed on different chromosomes (Figs. S3A–S3E), for example, six thiolase genes were scattered on four chromosomes in *D. melanogaster* (Fig. S3A), which is similar to thiolase genes in the human (*Anbazhagan et al., 2014*). However, different members of a certain class are often tandem distribution, such as *DmelSCP2 (type-1)-1* and *DmelSCP2 (type-1)-2* in *D. melanogaster* (Fig. S3A) and *BmorT1-1*, *BmorT1-2*, and *BmorT1-3* in *B. mori* (Fig. S3C). Meanwhile, we also detected the distribution of the 10 T1-thiolase genes in *P. xylostella* and 7 CT-thiolase genes in *A. pisum*, which were distributed on several small unassembled scaffolds. Whether they are distributed in tandem, we still need to wait for the scaffold sequences to be integrated into the corresponding chromosome in future. In general, tandem duplication might be the main mechanism for enlarging thiolase family in insects.

The syntenic relationships of thiolase genes were investigated among *B. mori*, *H. melpomene*, *D. melanogaster*, *T. castaneum*, and *A. mellifera* because their genome sequences have been assembled into chromosome levels. The results indicated that only four genes exhibited the syntenic relationships between *B. mori* and *H. melpomene*, that is, *BmorT2* and *HmelT2*, *BmorTFE* and *HmelTFE* (Fig. S3F). Interestingly, except for the tandemly duplicated genes, amount of thiolase genes often present orthologous relationships among insect, human, and *M. tuberculosis* (Fig. 1). It was suggested that thiolase family is an ancient gene family. Even in insects, the age of thiolase gene differentiation is relatively long. Thus, the discovery of fewer syntenic genes implies that thiolases might mainly locate in some non-conserved genomic blocks (*The Heliconius Genome Consortium, 2012*).

## Subcellular localization of thiolase proteins

Subcellular localization refers to the specific location of a certain protein or the expression product of a certain gene in the cell. Protein subcellular localization is closely related to protein functions (*Pereto, Lopez-Garcia & Moreira, 2005*; *Wang et al., 2014*). Only when the protein is positioned correctly can it perform normal biological functions. In this study, subcellular localization of all the 137 insect thiolase proteins was predicted by PSORT II server (https://www.genscript.com/tools/psort), which were cytosolic, mitochondrial, or peroxisomal enzymes (Fig. 3C; Fig. S2B). Generally, most of the TFE- and T2-thiolase proteins were located in the mitochondrion, T1- and CT-thiolases were cytosolic, and SCP2-thiolases were peroxisomal proteins. Previous studies suggested that the mitochondrial and peroxisomal thiolase proteins were mainly involved in the fatty acid $\beta$-oxidation pathway (*Pereto, Lopez-Garcia & Moreira, 2005*), and cytosolic localization was related to the biosynthesis of acetoacetyl-CoA (*Kursula et al., 2005*). However, in a certain class of thiolase, there are always a few exceptions to the cellular location in some species (Fig. 3C; Fig. S2B), which suggested that its function might have diverged during evolution.

## Conserved domain characteristics and catalytic residues

To identify the potential domains of insect thiolase proteins (Table S1), it was performed hmmscan analysis in Pfam database. The results indicated that all the thiolases contained Thiolase_N and Thiolase_C domains (Fig. 4A). In addition to the thiolase domains, SCP2-thioloase (type-1) has a typical sterol carrier protein 2 (SCP2) domain at C terminal. Unexpectedly, some of the members in TFE, CT, and T2 classes contained a ketoacyl-synt (beta-ketoacyl synthase) domain within Thiolase_N (Table S1). We carefully checked the alignments of hmmscan search. It was found that the *E* value was around the threshold 1e−5, and only about 50 amino acids can be aligned, which are much shorter than 250 amino acids of the ketoacyl-synt domain (Pfam ID, PF00109). Thus, thiolases may not contain the real ketoacyl-synt domain, and just show certain similarities with it (*Huang et al., 1998*). Therefore, based on the domain characteristics, all the insect thiolase encoding genes were classified as 2 groups (Fig. 4A).

The conserved sequence blocks of the 20 insects, humans, and *M. tuberculosis* were analyzed (Fig. S4 ; Fig. 4B). The CxS-motif is the most important sequence fingerprint in the N-terminal domain, which provides the nucleophilic cysteine (*Zeng & Li, 2004*; *Mazet et al., 2011*). Except for some incomplete sequences, almost all the thiolases contained the cysteine residue (Fig. S4 ; Fig. 4B). The histidine of the GHP-motif contributes to the oxyanion hole of the thioester oxygen (*Merilainen et al., 2009*). It was indicated that GHP-motif was highly conserved in insects, humans, and *M. tuberculosis* (Fig. S4 ; Fig. 4B). The cysteine of CxGGGxG-motif provides the catalytic residue of the active sites. Except for SCP2-thiolases, the catalytic cysteine was retained in almost all the other thiolases (Fig. S4). In addition, the asparagine side chain of the NEAF-motif interacts with important catalytic water (*Mazet et al., 2011*). However, NEAF-motif was replaced by HDCF-motif in all of the SCP2-thiolases (Fig. S4 ; Fig. 4B). Based on the comparison of the sequence fingerprints, it was indicated that the catalytic mechanisms of the insect thiolases might be similar to that of thiolases from mammals and bacteria.

## Molecular modeling of insect thiolases

In recent years, the crystal structures of some thiolases have been gradually resolved in bacteria, fish, and mammals (*Harijan et al., 2013*; *Kim et al., 2015*; *Xia et al., 2019*). The high sequence similarities (>60%) may help to build more accurate 3D structures for the insect thiolases (*Arnold et al., 2006*). Based on homology modeling using SWISS-MODEL Interactive Workspace, we found that thiolase sequences within a class were very conserved among different organisms. For instance, BmorTFE-thiolase and BmorSCP2-thiolase (type-1) shared 67.58% and 61.63% identities with its corresponding modeling templates from human (PDB ID: 6dv2.1.A) and zebrafish (6hrv.2.A), respectively. In this study, the modeling structures of some representative thiolases were presented (Figs. 5A–5F). The structural similarities of the monomeric forms were detected with magic fit in Swiss-PdbViewer (*Guex & Peitsch, 1997*). The RMSD values were ranged from 0.25 Å to 0.91 Å among BmorT2, BmorT1-1, DmelCT, and PxutAB-1, while BmorTFE and BmorSCP2 (type-1) shared from 1.12 Å to 2.01 Å with the others (Table S3). It was indicated that

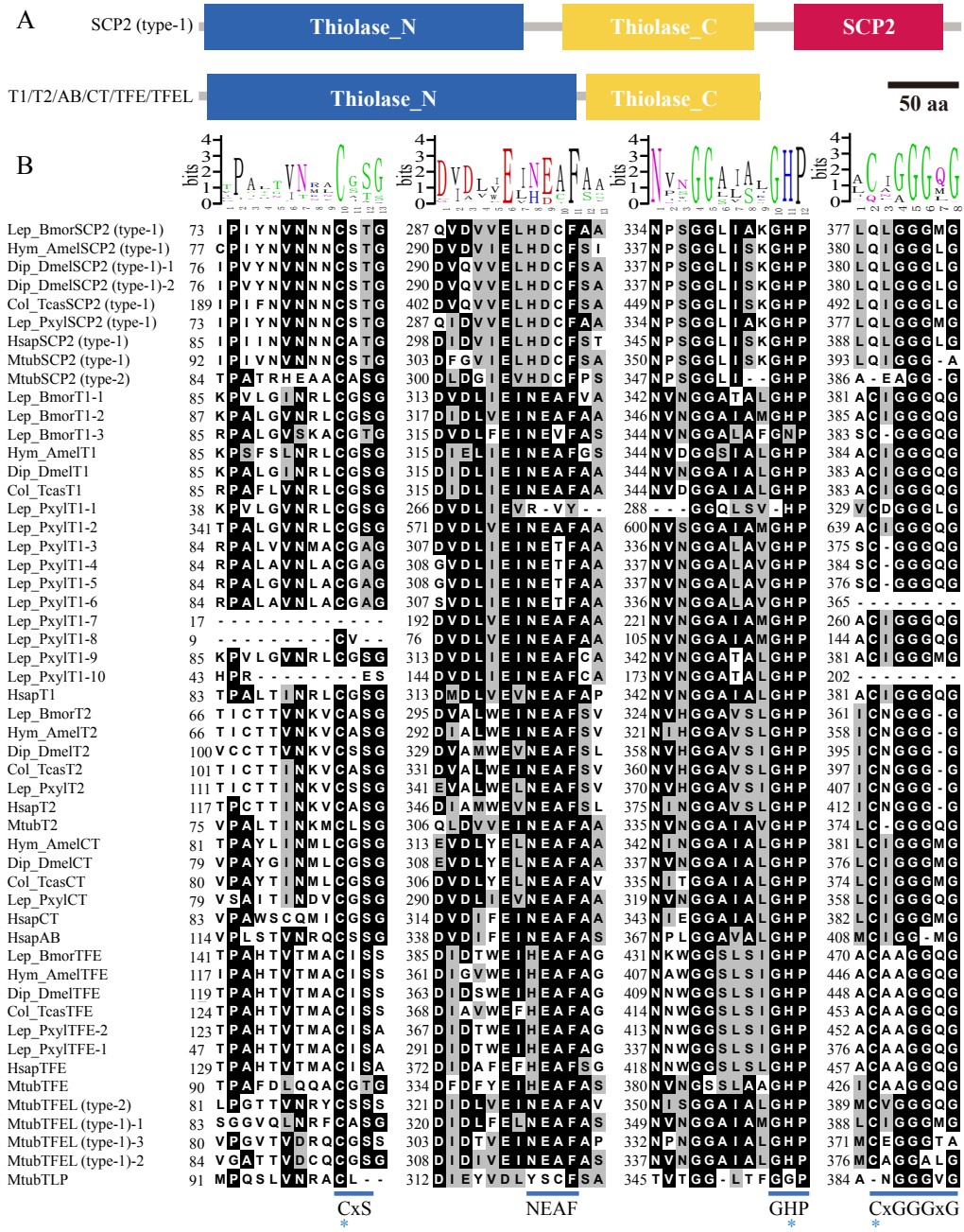

**Figure 4 Conserved domains and fingerprints of insect thiolases compared with humans and bacterium *M. tuberculosis*.** (A) The conserved domains predicted by hmmscan against Pfam database. Three types of domains were found in insect thiolases, including N-terminal of thiolase (Thiolase_N), C-terminal of thiolase (Thiolase_C), and sterol carrier protein 2 (SCP2). BmorSCP2 (type-1) and BmorT1-1 were used to represent the common structures, respectively. (B) Highly conserved sequence blocks containing fingerprints CxS, NEAF, GHP, and CxGGGxG. Three catalytic residues responsible for thiolase activity are indicated by stars. The alignement logos were generated by WebLogo (http://weblogo.berkeley.edu/logo.cgi).

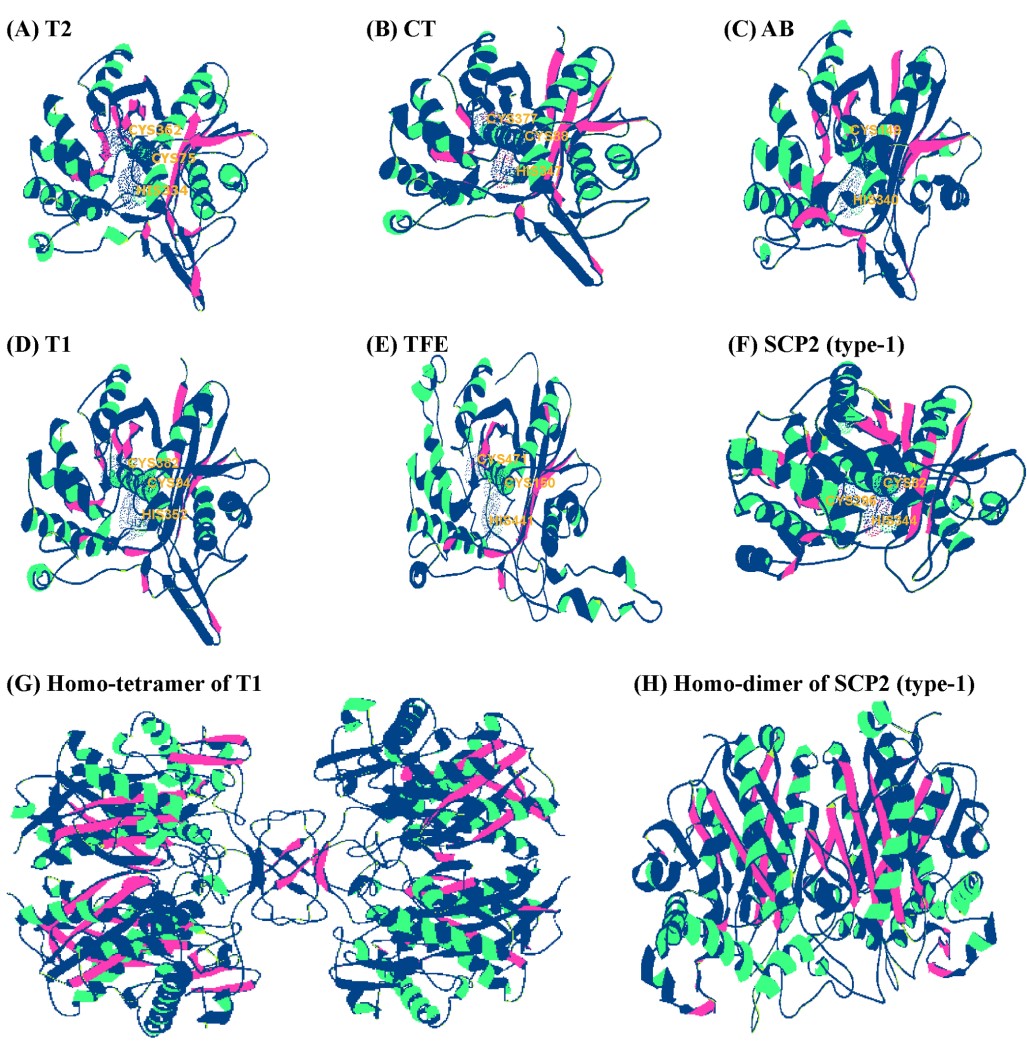

**Figure 5** **Protein homology modeling and 3D structures of insect thiolases.** The monomer structures of some representative thiolases were shown, including BmorT2 (A), DmelCT (B), PxutAB-1 (C), BmorT1-1 (D), BmorTFE (E), and BmorSCP2 (type-1) (F). The corresponding templates for homology modeling were PDB ID 6bjb.1.A, 4wyr.1.A, 1afw.1.A, 4wyr.1.A, 6dv2.1.A, and 6hrv.2.A, respectively. The dotted surface of the three catalytic residues was presented in (A) to (F) monomer structures. (G) Homo-tetramer of BmorT1-1. (H) Homo-dimer of BmorSCP2 (type-1).

T1, T2, CT, and AB classes share more similar 3D structures than TFE-thiolase and SCP2-thiolase (type-1) (Figs. 5A–5F). This phenomenon is widespread in both humans and *M. tuberculosis* (*Harijan et al., 2013*; *Anbazhagan et al., 2014*). For the quaternary structure, different thiolases also have certain differences. For example, BmorT1-1 and BmorSCP2 (type-1) were homo-tetramer and homo-dimer, respectively (Figs. 5G and 5F). The results of 3D structural modeling showed that different classes of thiolase genes still present some extent divergence in tertiary or quaternary structures.

SCP2-thiolase (type-1) was widely distributed in insects, mammals, and bacteria (Table 1). One single structural gene referred to as the sterol carrier protein x (*SCPx*) gene encodes

a full-length protein comprised of 3-oxoacyl-CoA thiolase (known as SCP2-thiolase) and sterol carrier protein 2 (*Seedorf et al., 1994*; *Gallegos et al., 2001*). The C-terminal SCP2-domain containing the peroxisomal targeting signal is needed for the targeting of full-length SCPx into the peroxisomes. The SCP2-thiolase and SCP2 protein are produced from SCPx via proteolytic cleavage by peroxisomal proteases (*Seedorf et al., 1994*). Based on the homology modeling, the tertiary and quaternary structures of mature SCP2-thiolase (type-1) protein were presented in Figs. 5F and 5H, respectively. For insect SCP2-thiolases, the canonical CxGGGxG-motif is also absent, and the NEAF-motif has been replaced by HDCF-motif (Fig. 4B). The previous studies indicated that HDCF-motif might provide the catalytic cysteine in bacteria, mammals, and fish (*Harijan et al., 2013*; *Kiema et al., 2019*). Based on the structural modeling, the cysteine of HDCF-motif is very close to the other two catalytic sites in protein spatial conformation (Fig. 5F). Therefore, the catalytic cysteine of the insect SCP2-thiolases might be not provided by CxGGGxG-motif but HDCF-motif.

## Expression profile and potential functional diversity

To understand the potential functional diversity of the insect thiolases, the silkworm, *B. mori*, was used as a model organism to perform expression profile analysis in the various tissues and sex pheromone glands (PGs) at different developmental stages. In the silkworm, genome-wide microarray with 22,987 oligonucleotides was designed and surveyed the gene expression profiles in multiple tissues on day 3 of the fifth-instar larvae (*Xia et al., 2007*). 5 out of 6 thiolase genes were found its corresponding probes (Fig. 6A). The microarray data indicated that *BmorSCP2 (type-1)*, *BmorT2*, *BmorTFE*, and *BmorT1-1* have expression signals at least one of the 9 tissues. Relatively, *BmorT2* and *BmorT1-1* showed ubiquitous expressions. Meanwhile, the expression profiles of the four genes were similar between females and males, respectively.

To validate the expression profiles of the silkworm thiolase genes, the mixed male and female tissues were used to perform RT-PCR validation on day 3 of the fifth-instar larvae (Fig. 6B). In total, 5 out of the 6 thiolase genes presented expression evidence. Relatively, *BmorT1-2* and *BmorTFE* showed predominant expressions in hemocyte and head, respectively (Fig. 6B), while *BmorT1-1* was widely expressed in various tissues. In addition, sex pheromone glands of different developmental stages were used to detect the expressions of thiolase genes (Fig. 6C). Four thiolase genes presented expression signals in the silkworm PGs. Relatively, *BmorT1-1* showed the highest expression on day 8 of pupae. *BmorTFE*, *BmorT2*, and *BmorSCP2 (type-1)* presented expressions at all the developmental stages. Interestingly, the expression levels of all three genes were declined in the mated female PGs (Fig. 6C). These expression analyses might help us understand the functional divergence of the thiolase genes in the silkworm.

## DISCUSSION

Thiolases are widely distributed in all organisms and are essential for a range of metabolic pathways. With the development of sequencing technology, it provides the possibility for us to identify and compare insect thiolase at the whole genome level. In this study, 137 thiolase genes were identified in the 20 representative species from 7 insect orders

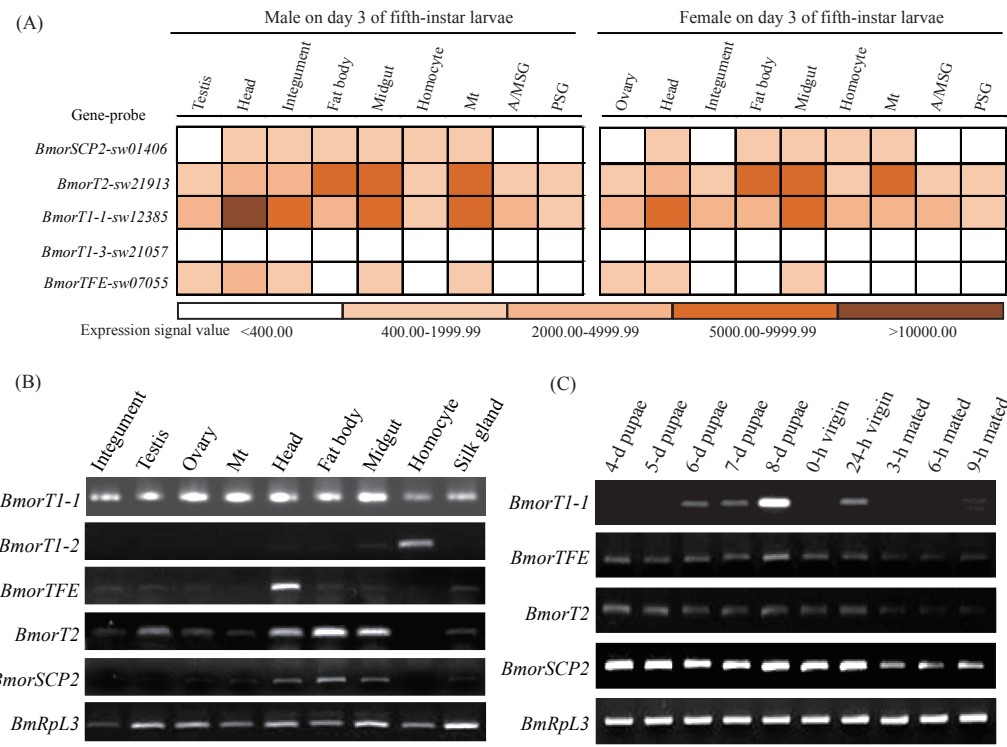

**Figure 6 Expression profiles of the thiolase genes in the silkworm.** (A) A chromatic scale diagram of expression levels in various tissues on day 3 of the fifth-instar larvae. The expression signal values were from the silkworm microarray, and signal values <400 were recognized no expression (*Xia et al., 2007*). (B) Expression patterns of the thiolase genes in the various tissues of the fifth-instar larvae. Expression signal of *BmorT1-3* was not detected. It was not presented in the figure. (C) Expression profiles in the sex pheromone glands at different developmental stages of females. Expressions of *BmorT1-2* and *BmorT1-3* were not detectable in the PGs. 0-h and 24-h vergin: 0-h and 24-h old vergin adults after eclosion ; 3-h, 6-h and 9-h mated: female moths mated 3 h, 6 h and 9 h.

(Table 1; Table S1). The insect thiolases were mainly classified into five classes, including CT-thiolase, T1-thiolase, T2-thiolase, TFE-thiolase, and SCP2-thiolase. It was indicated that *P. xylostella*, *A. pisum*, and *P. xuthus* showed more duplications, resulting in a total number of 15, 12, and 10 genes, respectively. *Z. nevadensis* and *H. melpomene* have the least number of genes (Table 1). In addition to a certain differentiation in the number of genes, Thiolase_N or Thiolase_C domains of 9 thiolase genes were missing (Table S1). It is worth noting that the quality of the genome may have a certain impact on the number of genes and the integrity of gene structures. Whether the incomplete thiolase genes were pseudogenes or not (Table S1) needs further verification by the high-quality genome in the future.

Two groups of thiolases were identified in animals: 3-oxoacyl-CoA thiolase and acetoacetyl-CoA thiolase, which participates in different catabolic (fatty acid oxidation and bile acid formation) and anabolic (cholesterogenesis, ketone body synthesis, fatty acid elongation) processes (*Antonenkov, Veldhoven & Mannaerts, 1999*). It is well known that cholesterol is a precursor of molting hormone, 20-hydroxyecdysone (20E), and is a

structural component of cell membranes (*Gilbert, Rybczynski & Warren, 2002*). Due to the lack of squalene monooxygenase and lanosterol synthase for the synthesis of cholesterol, insects can not autonomously synthesize the 20E precursor (*Guo et al., 2009*). Alternatively, insects can obtain cholesterol or other sterols from their diet to meet the needs of growth and development. In *Spodoptera litura*, one sterol carrier protein x (*SCPx*) gene encoding a sterol carrier protein 2 and a 3-oxoacyl-CoA thiolase known as SCP2-thiolase (type-1) showed predominant expression in the midgut, and its coding SCP2 was involved in the absorption and transport of cholesterol (*Guo et al., 2009*). In the silkworm, *SCPx* gene has been cloned (*Gong et al., 2006*). It presented expressions in the midgut, fat body, and head on day 3 of the fifth-instar larvae in the silkworm (Fig. 6B), which suggested that the SCP2 protein might have a similar function with that of *S. litura* (*Guo et al., 2009*). More important, the SCP2-thiolase (type-1) encoded by *SCPx* plays a crucial role in the oxidation of the branched side chain of cholesterol to form bile acids in vertebrates (*Ferdinandusse et al., 2000*), while the physiological role has not been characterized in insects. Fortunately, the expression of the SCP2-thiolase (type-1) has also been detected in the prothoracic glands of *Spodoptera littoralis*, which are the main tissue producing the insect molting hormone (*Takeuchi et al., 2004*). Thus, whether SCP2-thiolase (type-1) of the silkworm and other insects play role in the oxidation of cholesterol and participates in ecdysone synthesis needs further study.

In insects, juvenile hormone (JH) is an important regulator for growth and development (*Kinjoh et al., 2007*) and several thiolases have been cloned and suggested to be related to JH biosynthesis (*Kinjoh et al., 2007*; *Zhu et al., 2016*; *Zhang et al., 2017*). Acetoacetyl-CoA thiolase catalyzes two molecules of acetyl-CoA to form acetoacetyl-CoA, which is the first enzyme in JH biosynthesis (*Kinjoh et al., 2007*; *Zhang et al., 2017*). The candidate acetoacetyl-CoA thiolases related to JH biosynthesis were cloned in *B. mori* and *Helicoverpa armigera* (*Kinjoh et al., 2007*; *Zhang et al., 2017*). In this study, those two acetoacetyl-CoA thiolase genes were classified as T2-thiolases (BmorT2 and HarmT2), and they shared high sequence identities with the other T2-thiolases (Table S4). For example, BmorT2-thiolase shared 82.71% sequence identity with HarmT2. In *H. armigera*, temporal expressions of *HarmT2-thiolase* keep pace with JH fluctuations, and its expression can be inhibited by a juvenile hormone analog (*Zhang et al., 2017*). The expression of *BmorT2-thiolase* was relatively abundant in the head where the JH synthetic gland, corpora allata (CA), is located (Figs. 6A and 6B). Interestingly, we found BmorTFE- and BmorT1-1-thiolase also showed high expressions in the larval head (Fig. 6A). In humans, TFE- and T1-thiolases catalyze thiolytic cleavage of 3-ketoacyl-CoA into acetyl-CoA and acyl-CoA (*Anbazhagan et al., 2014*; *Xia et al., 2019*). However, T1-thiolase has been found synthetic and degradative activities in *Ostrinia scapulalis* (Lepidoptera: Crambidae). Therefore, whether T2-, TFE- and T1-thiolases were involved in JH biosynthesis is still worthy of experimental validation.

Acetyl-CoA is often used as the initial precursor for sex pheromone biosynthesis in insects (*Matsumoto, 2010*). Degradative thiolases may supplement with sufficient acetyl-CoA for sex pheromone synthesis (*Brabcova et al., 2015*). In this study, expression profiles of the thiolase genes were detected in the sex pheromone glands at different developmental stages in the silkworm (Fig. 6C). Relatively, *BmorSCP2 (type-1)* maintains a high level of

expression in the PGs on day 4 of pupae to 24-h-old virgin female moth. However, its expression level was sharply declined in the mated female PGs (Fig. 6C). The previous study suggested that an over 6-h mating duration can terminate the sex pheromone production in the silkworm (*Ando et al., 1996*). The expression pattern of *BmorSCP2 (type-1)* was consistent with sex pheromone production (*Matsumoto, 2010*), which suggested that it might be involved in sex pheromone biosynthesis. Generally, it is tempting to assume that a thiolase expressed in a specific tissue might obtain a specific role. Thus, the functional diversification and physiological roles of insect thiolases need yet further experimental validation.

## CONCLUSIONS

In the present study, genome-wide identification of the thiolase gene family was conducted in multiple insect genomes. A total of 137 thiolase genes were identified in 20 insects from seven orders. About 80% of the thiolase genes have two or more introns, and its exon/intron structures reserve diversification. Based on the prediction, all the thiolase proteins are located in the mitochondria, cytosol, or peroxisome, and thiolases of the same class often have similar cellular localization. Four highly conserved sequence fingerprints were found in the insect thiolase proteins, including CxS-, NEAF-, GHP-, and CxGGGxG-motifs. Homology modeling analysis indicated that 3D structures of the insect thiolases share similar to mammals, fishes, and microorganisms. Expression pattern analysis suggested some thiolase genes may be involved in steroid metabolism, JH, and sex pheromone biosynthesis pathways in *B. mori*. These results might provide valuable information for the functional exploration of thiolase proteins in insects.

## ACKNOWLEDGEMENTS

The author sincerely thanks the anonymous reviewers for their affirmation and constructive comments on the manuscript.

### Funding

This study was supported by the Initiation Fund (No. 15E022) and the Teaching Reform Research Project (No. Jgxmyb18151) of China West Normal University. The funders had no role in study design, data collection and analysis, decision to publish, or preparation of the manuscript.

### Grant Disclosures

The following grant information was disclosed by the author:
Initiation Fund: 15E022.
China West Normal University:  Jgxmyb18151.

### Competing Interests

The authors declare there are no competing interests.

## Author Contributions

- Shou-Min Fang conceived and designed the experiments, performed the experiments, analyzed the data, prepared figures and/or tables, authored or reviewed drafts of the paper, and approved the final draft.

## Data Availability

The raw data and the uncropped blots are available as Supplemental Files.

## Supplemental Information

Supplemental information for this article can be found online at http://dx.doi.org/10.7717/peerj.10393#supplemental-information.

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
