# Peer review of "Genome-wide identification and analysis of the thiolase family in insects"

_PeerJ, doi:10.7717/peerj.10393_

## Round 0.1 · original submission · Major Revisions

Your manuscript has been reviewed by three experts in the field. As you can see from their comments below, all of them admit the value of your work; one even recommends its acceptance as it is. However, I think that manuscript should be revised following their comments by the remaining two reviewers (mainly Reviewer 2). Please read their comments carefully and revise the manuscript accordingly. Looking forward to your revised manuscript.

Reviewer 1 ·

Basic reporting

no comment

Experimental design

No comment

Validity of the findings

No comment

Additional comments

This research focused on thiolase family in insects. Thiolases plays important roles in lipid metabolism but it is poorly understood in insects. This study identifies the genes and provides key information for the future function analysis. The manuscript is generally written well. I have some comments as following:
1、Line 164, it is better to mention somewhere genome quality potentially influences the results, like the number of genes identified and the deduced evolutionary process.
2、Line 178, the author can specify which group have no intron.
3、Line 181, this part reads like introns can help genes respond to enviromental challenges, which is contradicting to the first part of the sentence.
4、Line 182-183, the findings/results cannot support this discussion/opinion yet.
5、Line 204, reference.
6、Line 243, the author should list protein identifies between target protein and human template for all proteins. But the author only give the information for TFE and SCP2.

Reviewer 2 ·

Basic reporting

The author used clear and unambiguous English, but there are several simple grammatical and typographical errors to be corrected before publication, which are listed below.

Lines 21, 311: “. While, ” => “, while ”
Line 31: Italicize species name.
Line 44: Fig S1A => Fig. S1A
Line 62: belonged => belong
Lines 124–125: Use singular form after “every “
Line 247: We => we
Line 267: contained => containing
Line 273: “Therefore ” => “Therefore, ”
Line 338: delined => declined

There are also several texts and words which should be improved for clarity and better understanding.
Line 90: “gene models” may be “genes”?
Lines 154–155: The sentence, “In total, …”, should be improved into a clearly understandable text.
Line 222: “Except for” here may confuse readers. It may be better to use “In addition to” instead.
Line 224: “seem to be contained” => “contained”
Line 227: largely => much
Line 269: “SCPx by proteolytic cleavage via peroxisomal” => “SCPx via proteolytic cleavage by peroxisomal”
Line 350: and => or

Some figures and table seem to be in insufficient quality:
Fig. 1: The visibility of dots on the nodes is poor.
Figs. 3B and S2A: Scale bars? cannot be read and look strange.
Fig. 5: The resolution of some figures seems to be low. Numbers with dotted surfaces are shown, but hard to recognize.
Fig. 6: (A) The change of the color for expression <400 into white is recommended for clarity. (B) and (C) The legends seem to be wrong texts.
Table 1: Double-byte spaces appear in Iso_Z. nevadensis line and should be removed.

Titles of supplemental figures are missing and should be placed. In addition, Figs S2 and S3 may also need legends for better understanding.

Experimental design

There are several concerns about details of the methods listed below.

Line 91: The URL of SilkDB (http://silkworm.genomics.org.cn) cannot be accessed.
Line 101, 153: The detail of the known sequences should be mentioned, probably by using Table S2.
Line 102: The condition of BLASTP search should be written.
Lines 108–109: The detail of how did the author validate “The validated thiolases” should be clearly written.
Line 123: The author should specify the species used as “three model species” here or in the result texts and explain why selected them.
Line 135: The author should specify the kind of “similarity”. Is it sequence similarity?
Lines 137–138: “Using BmorT2 as the reference…”. I cannot understand where this was used.
Lines 209–210: The method for “In this study, subcellular localization of all the 137 insect thiolase proteins was predicted” was not written.

Validity of the findings

Several sentences in the conclusion seem to be exaggerated or over generalized and should be improved.
Line 349: “randomly and unevenly distributed” has weak evidence derived from small number of samples.
Lines 350–351, 354–355: The results are based on prediction.
Lines 353–354: “3D structures were highly conserved between insects and other vertebrates” was not clearly shown in general.

Additional comments

The author identified thiolase genes in 20 insects and analyzed their features about phylogeny, genes, 3D-structures, expressions of proteins, etc.

Major concerns:
* The analyses seem to be well done, but some of them were not comprehensively done: for example, chromosome distribution and conserved domain analysis.

* The manuscript needs more discussion “based on the results”, especially for the expression profile parts and should be largely improved.

* Some results are discussed only in insects thiolases (e.g., Lines 190–194, 254–255). The author should evaluate the results by comparing them with the known thiolases of the other organisms. Whether the results are insects-specific or not is important for better understanding the thiolases of insects.

* Fragment sequences like only N-terminal parts are treated same as whole genes. Those sequences may be pseudogenes and should be separately treated. In addition, as the author also mentioned in the text, some of the full-length genes lack the catalytic residues important for thiolases and may lose their functions as thiolases. Those incomplete genes should be distinguished with likely genuine genes (e.g., in Table 1).

Minor concerns:
Lines 185–187: I couldn’t understand why the sentence, “It seems that ...”, was derived. Maybe, more explanation is needed here.
Line 251: The values of sequence similarity are better to be shown. This sentence may be better to appear in line 246.
Line 255: The values of 3D structural similarity are better to be shown.
Lines 258–260: “However, …” The meaning of this sentence is hard to understand for me. Maybe more explanation is needed.
Line 264: How did the author judge “this type of thiolase is the most divergent”?
Lines 271–276: The order of sentences may be better to be reordered: known knowledge as first.
Lines 324: “relatively high enriched” seems to be not appropriate phrase for this.
Lines 325–326: “It further supports that insect T2-thiolase genes may be involved in the JH synthesis pathway.” Generalization as “insect T2-thiolase genes“ seems to be unreasonable.

Reviewer 3 ·

Basic reporting

In this manuscript, the author (Shou-Min Fang) identified thiolase genes in 20 representative insect genomes using bioinformatics analysis. Insect thiolases were mainly classified into five classes (CT-thiolase, T1-thiolase, T2-thiolase, TFE-thiolase, and SCP2-thiolase). Their intron numbers and exon/intron structures, and predicted subcellular localization are variable. Homology modeling indicated that insect thiolases share similar 3D structures with mammals and other vertebrates. Transcription analysis suggested that, in B. mori, some thiolases may be involved in steroid metabolism, juvenile hormone and sex pheromone biosynthesis. In general, the manuscript is well written and organized.

Experimental design

The manuscript is pretty well organized and the experiment is carefully designed. Identification of thiolase genes in sequenced insect genomes provide a useful resource for further functional studies.

Validity of the findings

no comment

---

## Round 0.2 · Minor Revisions

Your revised manuscript has been reviewed by the same two reviewers, whose comments are shown below. As you will see in their comments, one of them now recommends its acceptance as is while the other still raises several points. Please read these comments carefully and re-revise the manuscript, unless you disagree with them. Particularly, you should clarify why you have used only four out of 20 insects data.

Reviewer 1 ·

Basic reporting

No comment.

Experimental design

No comment.

Validity of the findings

No comment.

Additional comments

The author has addressed my concerns well, and I have no further comment.

Reviewer 2 ·

Basic reporting

The author has corrected most of grammatical and typographical errors, and problems of figures and tables, I previously commented, but there are still some are left to be corrected and new ones appeared:

Line 108: have => “that have” or “having”; Thionlase => Thiolase; recogized => recognized
Line 124: “, which” => because
Line 126: genes => gene
Line 236: “Unexceptly” may be “Unexpectedly”?
Line 258: higher => high
Line 304: “head and hemocyte” => “hemocyte and head”
Lines 304–305: “. While, ” => “, while ”
Line 338: “was suggested” => “suggested”
Line 344: “Spodopera littoralis” => “Spodoptera littoralis”
Line 355: higher => high
Line 390: suggests => suggested
The title of Table S2 in the word file is Table S1.

Experimental design

The author has adequately addressed all my comments.

Validity of the findings

The author has adequately addressed all my comments.

Additional comments

The author has addressed most of my comments. However, there are still some concerns that need to be resolved.

Major concerns:
* The author should use all the insects thiolases in the conserved sequence block analysis. In the present manuscript, only 4 of 20 insects were used to derive the conclusion for insects thiolases, which is unreasonable.

* The author seems to regard BmorT2-thiolase as involving JH synthesis pathway only based on weak estimation from gene expression and it still needs further experimental validations similar to those for HarmT2-thiolase by (Zhang et al. 2017).
** Lines 360–361: “Thus, insect T2-thiolases may be involved in the JH synthesis pathway.”: I think it is unreasonable to generalize the JH synthesis about insects from only an example in H. armigera and an estimation in B. mori.
** Lines 365–366: “except for T2-thiolase” is not appropriate, I think.

Minor concerns:
Lines 266–270: Why did the author use only BmorT2 as the reference to calculate the RMSD values here? It is not enough result to support the description: “It was indicated that T1, T2, CT, and AB classes share more similar 3D structures than TFE-thiolase and SCP2-thiolase (type-1)”.

Line 270: “This phenomenon is widespread in both humans and M. tuberculosis” needs references.

Lines 270–274: The sentences, “Generally, T1-, T2-, ... biological unit (Xia et al. 2019)”, seem to mostly describe the general fact and are not appropriate in the Results section. The author should better describe what was derived from the results of Figs. 5G and 5H, or remove this part.

Lines 303, 307: “In total, 5 out of the 6 thiolase genes presented expression evidence.”, “Four thiolase genes presented expression signals in the silkworm PGs.”: Figs. 6B and 6C do not show the results of all the 6 thiolase genes. The author should show all the results in the figure.

Lines 337–339: “It” in the sentence, “It presented expressions ...”, indicates SCPx gene, but Fig.6B shows the expression result of SCP2 thiolase gene. It is better to fill the gap between SCPx gene, SCP2 thiolase gene/protein, and SCP2 protein just before here for easy understanding.

Lines 348–350: The two sentences may be better to be combined into one sentence.

Lines 376–377: The author wrote that “Simultaneously, BmorTFE and BmorT2 showed similar expression patterns as BmorSCP2 (type-1).”, but BmorTFE and BmorT2 seem to show different expression patterns from BmorSCP2 (type-1) in Fig. 6C for me.

Lines 31–33, 390–391: “may be involved” or “might be involved” is appropriate instead of “were involved” because of the weak evidences.

---

## Round 0.3 · Minor Revisions

Your revised manuscript has been reviewed by the same reviewer who made some comments in its previous version. This time, the reviewer gives only a few minor comments. I believe that I can accept the manuscript in the next revision without another round of review.

Reviewer 2 ·

Basic reporting

There are two new errors appeared in this revision:

Line 249: “in in”=> “in”
Line 272: structrue => structure

Experimental design

No comment.

Validity of the findings

No comment.

Additional comments

The author has addressed all my comments.
Below is just a recommendation for easy understanding.

The omission of data with no signals in Figs. 6B and 6C is better to be explicitly written in the main manuscript and/or the figure legend.

---

## Round 0.4 · accepted · Accept

As I confirm that you addressed all the points raised by the reviewer, I now decide to accept your manuscript in PeerJ.